# Lasp1 Expression Is Implicated in Embryonic Development of Zebrafish

**DOI:** 10.3390/genes14010035

**Published:** 2022-12-22

**Authors:** Ilaria Grossi, Marco Schiavone, Elena Cannone, Oana Andreea Grejdan, Chiara Tobia, Francesca Bonomini, Rita Rezzani, Alessandro Salvi, Giuseppina De Petro

**Affiliations:** 1Department of Molecular and Translational Medicine, Division of Biology and Genetics, University of Brescia, 25123 Brescia, Italy; 2Department of Molecular and Translational Medicine, Division of Experimental Oncology and Immunology, University of Brescia, 25123 Brescia, Italy; 3Department of Clinical and Experimental Sciences, Division of Anatomy and Physiopathology, University of Brescia, 25123 Brescia, Italy

**Keywords:** Lasp1 expression, zebrafish model, embryonic development, apoptosis

## Abstract

The LIM and SH3 domain protein 1 (LASP1) was originally identified in metastatic breast cancer and mainly characterized as a cytoskeleton protein overexpressed in various cancer types. At present, little is known about LASP1 expression in physiological conditions, and its function during embryonic development has not been elucidated. Here, we focused on Lasp1 and embryonic development, choosing zebrafish as a vertebrate model. For the first time, we identified and determined the expression of Lasp1 protein at various stages of development, at 48 and 72 h post-fertilization (hpf), at 6 days pf and in different organs of zebrafish adults by Western blotting, 3D light-sheet microscopy and fluorescent immunohistochemistry. Further, we showed that specific *lasp1* morpholino (MO) led to (i) abnormal morphants with alterations in several organs, (ii) effective knockdown of endogenous Lasp1 protein and (iii) an increase in *lasp1* mRNA, as detected by ddPCR. The co-injection of *lasp1* mRNA with *lasp1* MO partially rescued morphant phenotypes, thus confirming the specificity of the MO oligonucleotide-induced defects. We also detected an increase in apoptosis following *lasp1* MO treatment. Our results suggest a significant role for Lasp1 in embryonic development, highlighting zebrafish as a vertebrate model suitable for studying Lasp1 function in developmental biology and organogenesis.

## 1. Introduction

In 1995, Tomasetto et al. originally identified the LIM and SH3 protein 1 (LASP1) in a c-DNA library of human breast cancer metastatic lymph nodes [1]. The human *LASP1* gene, located on chromosome 17q11-21.3, encodes a polypeptide chain of 261 amino acids originally characterized as a structural cytoskeletal protein (https://www.genecards.org/cgi-bin/carddisp.pl?gene=LASP1, accessed on 10 October 2022). Since then, several authors have studied LASP1 expression, at the mRNA or protein level, in different cancer types [2,3,4,5,6,7,8], reporting that its overexpression is inversely correlated with poor prognosis in breast and prostate cancer, medulloblastoma and hepatocellular carcinoma (HCC). Concerning HCC, we have previously shown that *LASP1* mRNA is significantly overexpressed in HCC tissues, mainly in HCC developed in cirrhotic liver, and that levels of LASP1 protein and mRNA expression are comparable and increase in recurrent HCC [9]. Further, several studies have shown that LASP1 plays an important role in tumour development and metastases, and RNAi knock-down of LASP1 has led to strong inhibition of proliferation and migration in various cancer cells. Currently, it is known that LASP1 is a multidomain protein that interacts with various proteins, that it is able to exert different roles in cell signalling and in transcriptional regulation, and that it is also able to activate survival and proliferation pathways in different cancer types [10]. In physiological conditions, LASP1 is ubiquitously expressed at low levels in normal human tissues (except smooth muscle) and is highly expressed in the hematopoietic system. Data reported in 2015 by Orth et al. [11] clearly show: (i) that *LASP1* mRNA is expressed at very distinct levels within non-neuronal normal tissues but at rather similar levels in various neuronal tissues; and (ii) that *LASP1* mRNA is expressed at very high levels in fetal brain and liver. Some authors have suggested a prominent role for LASP1 in fetal development [10], and, recently, others have shown that LASP1 expression in the rat hippocampus early in development is maintained throughout adulthood [12]. Therefore, in the current work, we focused on LASP1 expression and developmental biology and aimed to study the role of Lasp1 in a zebrafish animal model [13] with the purpose of investigating its role in development and organogenesis. We verified Lasp1 expression at various stages of zebrafish embryonic development and in adult zebrafish tissues, and we downmodulated Lasp1 expression by the morpholino (MO) approach, monitoring the effects on the morphants and on apoptosis. Our results highlight zebrafish as a suitable vertebrate model for studying Lasp1 function.

## 2. Materials and Methods

### 2.1. Zebrafish Maintenance

The zebrafish AB wild-type strain was maintained at the Facility of the University of Brescia at 28.5 °C in aerated saline water, under a 14 h light–10 h dark cycle, according to standard protocols [14]. For mating, males and females were separated in the late afternoon and the next morning were freed to start courtship, which ended with egg deposition and fertilization. Eggs were collected and maintained at 28.5 °C in fish water (0.5 mM NaH_2_PO_4_, 0.5 mM NaHPO_4_, 0.2 mg/L methylene blue, 3 mg/L instant ocean) in a Petri dish.

### 2.2. Protein Extraction and Western blotting

Zebrafish embryos at 24 and 48 h post-fertilization (hpf) and zebrafish early larvae at 72 hpf and at 6 dpf were deyolked by pipetting with a 200 µL pipette tip in cold Ringer’s solution, followed by centrifugation at 5000 rpm for 3 min at 4 °C. The deyolked embryos, larvae and tissues from adult zebrafish (including muscle, eye, brain, liver, intestine, and gastrointestinal tract tissues) were mechanically homogenized in cold RIPA buffer (50 mM Tris-HCl, pH 7.4; 150 mM NaCl; 1% NP40; 0.1% SDS; 2 mM EDTA; 0.5% Na-deoxycholate) with fresh protease inhibitors. The lysates were clarified by centrifugation at 14,000 rpm for 30 min at 4 °C, the supernatant was collected and the protein concentration was determined by the Bradford method. The cell extract from the human hepatocellular carcinoma cells, HA22T/VGH, was collected in 0.05% SDS and used as a positive control for LASP1 protein expression [9]. Constant amounts of proteins were loaded on 4–12% Novex NuPAGE Bis/Tris gels (Thermo Fisher Scientific, Inc., Waltham, MA, USA) under reducing conditions and electro-transferred onto nitrocellulose membranes. The following primary antibodies were used: mouse anti-human LASP1 monoclonal antibody (1:2000 in 0.3% BSA; clone 8C6 Millipore) and rabbit anti-human GAPDH (1:5000 in 0.3% BSA; GeneTex, Irvine, CA, USA). Primary antibodies were stained using HPR-coupled anti-mouse or anti-rabbit IgG (1:7500 in 0.3% BSA; Promega, Madison, WI, USA). Chemiluminescent signals were detected using an ECL Western blotting Substrate kit (Promega) and analyzed using ImageJ software (https://imagej.nih.gov) (accessed on 9 December 2022) [15].

### 2.3. Whole-Mount Immunofluorescence

Zebrafish specimens at 72 hpf were fixed in fresh 4% paraformaldehyde (PFA) in PBS for 4 h at RT and dehydrated in 100% methanol for storage at −20 °C. The embryos were rehydrated with an increasing scale of PBS/methanol and transferred into 100% PBS. After three washes with PBS-Tween 20 (0.1%) (PBS-T), the embryos were permeabilized using Proteinase K (10 μg/mL), re-fixed in 4% PFA and incubated in pre-blocking solution (PBS-T with 1% DMSO) for 20 min at RT. The samples were then incubated overnight at 4 °C in a blocking solution (PBS-T with 2% BSA, 2% heat-inactivated fetal bovine serum (FBS) and 1% DMSO). The embryos were incubated for 48 h at 4 °C in primary antibody (rabbit anti-human LASP1 polyclonal antibody, Millipore) diluted 1:100 in blocking solution, followed by incubation overnight at 4 °C with fluorescent secondary antibody (anti-rabbit Alexa 488, Thermo Fisher Scientific, Inc., Waltham, MA, USA) diluted 1:1000 in blocking solution [16]. Embryos were placed in 0.8% low-melting agarose (Top Vision Low Melting Point Agarose, Thermo Fisher Scientific, Inc., Waltham, MA, USA) and then put into the instrument holder. Images were acquired using a Zeiss 3D LightSheet microscope Z1 supported by ZenPro software and analyzed using FIJI imaging software.

### 2.4. Fluorescent Immunohistochemistry and Hematoxylin and Eosin Staining

Adult zebrafish were first anesthetized and sacrificed using 0.1 μM tricaine (Merck KGaA, Darmstadt, Germany) and dissected to obtain the brains, eyes, gastrointestinal tracts, livers, and muscles. The selected organs were embedded in OCT (optimum cutting temperature) compound, frozen using liquid nitrogen and isopentane, and sectioned using a cryostat. Sections 10 μm in thickness were adhered to poly-L-lysine-coated glass slides and stored at −80 °C until use. For immunofluorescence analysis, the sections were incubated at RT for 20 min in PBS-Tween 20 (0.1%) (PBS-T) with 1% DMSO. The sections were then incubated for 1 h in a blocking solution (PBS-T with 2% BSA, 2% heat-inactivated FBS and 1% DMSO). The sections were incubated overnight at 4 °C in primary antibody (rabbit anti-human LASP-1 polyclonal antibody, Millipore) diluted 1:100 in the blocking solution. The sections were washed three times with PBS-T containing 1% DMSO, followed by incubation for 2 h at RT with fluorescent secondary antibody (anti-rabbit Alexa 488, Invitrogen) diluted in the blocking solution. A negative control was obtained by incubating the sections without primary antibody and with fluorescent secondary antibody (Appendix A). For nucleus visualizations, the sections were washed three times with PBS-T and stained with DAPI (1:3000, Merck, Inc., Darmstadt, Germany) for 10 min. After three washes with PBS-T, the sections were mounted with Vectashield mounting medium (Vector Labs, Newark, CA, USA) and visualized using a Leitz fluorescence microscope. For histopathological evaluation, sections were stained with hematoxylin and eosin according to the standard protocol and observed with an optical light microscope (Olympus, Tokyo, Japan) at final magnifications of 4×, 10× and 20× [17,18].

### 2.5. Morpholino Injection

To knock-down Lasp1 expression, zebrafish embryos were injected with a mixture of specific morpholinos (*lasp1* MO) targeting the translation initiation sites of zebrafish *lasp1* transcript variants (5′-TGCTACACAGCGGGTTCATTTTGGA-3′ for *lasp1*-201; 5′-GCATCCTTTATGCCAGTACATATTC-3′ for *lasp1*-202) (Appendix A). A standard morpholino (STD MO) (5′-CCTCTTACCTCAGTTACAATTTATA-3′) was injected as a control, and p53 morpholino (p53 MO) (5′-GCGCCATTGCTTTGCAAGAATTG-3′) was co-injected with *lasp1* MO to exclude off-target effects due to MO-induced p53 activation. All MOs were designed and supplied by Gene Tools LLC, Philomat, Oregon, USA. Morpholinos were dissolved in fish water (0.5 mM NaH_2_PO_4_, 0.5 mM NaHPO_4_, 0.2 mg/L methylene blue, 3 mg/L instant ocean) and injected into 1-cell-stage embryos at doses of 1 μM, 10 μM and 100 μM. Three biological replicates were used for each assay. The evaluation of phenotypic effects and isolation of RNA and proteins were carried out at 48 hpf [13].

### 2.6. RNA Isolation and Droplet Digital PCR (ddPCR)

Total RNA was extracted with Trizol (Invitrogen) from morpholino-injected embryos. The cDNA was synthesized from 100 ng of total RNA using a First-strand cDNA synthesis kit (Thermo Fisher Scientific, Inc., Waltham, MA, USA), according to the manufacturer’s instructions. The reverse transcription (RT) reaction was performed at 37 °C for 50 min, followed by inactivation at 70 °C for 15 min in a T100 Thermal Cycler (Bio-Rad Laboratories). The synthesized cDNA was used as a template for the ddPCR experiments using the QX200 Droplet Digital PCR (ddPCR) System (Bio-Rad Laboratories, Inc., Hercules, CA, USA), and ddPCR was performed according to the ddPCR Supermix for Probes (Bio-Rad Laboratories) protocol, as previously described [19,20]. In brief, 2 μL of the resulting cDNA was prepared for amplification in a 20 μL reaction volume containing 2X ddPCR Supermix for Probes (Bio-Rad Laboratories), 20X TaqMan (Thermo Fisher Scientific) specific *lasp1* PCR probe assay (assay ID Dr03439091_m1) (Appendix A) and water. A no-template control (NTC) and a negative control for each reverse transcription reaction (RT-neg) were included. Concentration data for *lasp1* levels were obtained using QuantaSoft Software (Bio-Rad Laboratories) as copies/μL. The levels of *lasp1* were reported in percentages as the means of copies/μL measured in three ddPCR experiments and referred to the MO STD samples.

### 2.7. In Vitro mRNA Synthesis for the Rescue Experiments

The full-length coding region (CDS) of human *LASP1* was previously cloned into the pcDNA3.1-CT-GFP-TOPO vector (Thermo Fisher Scientific, Inc., Waltham, MA, USA), as reported by Salvi et al. [9]. The resultant construct, pGFP-LASP1, was linearized by NarI (Promega) and used as the template for the in vitro transcription of *LASP1* mRNA using a MEGAscript kit (Life Technologies), following the manufacturer’s instructions. The transcribed mRNA was purified using NucAway Spin Columns (Thermo Fisher Scientific, Inc., Waltham, MA, USA) and quantified using Nanodrop Lite. A mixture of *lasp1* MO and synthesized *LASP1* mRNA (50 ng) was used for the rescue experiments.

### 2.8. Acridine Orange Assay for Apoptosis Evaluation

Cell death in whole zebrafish embryos was measured with vital dye acridine orange staining at 48 hpf. Live embryos were immersed in 5 μg/mL acridine orange (Merck KGaA, Darmstadt, Germany) dissolved in PBS, in the dark for 1 h. Treatment with 1 µM rotenone (Merck KGaA, Darmstadt, Germany) was used as a positive control for cell death. Briefly, zebrafish embryos were treated at 46 hpf with 1 µM rotenone for 2 h and then incubated with acridine orange, as described for morphants..

After 3 × 10′ washes with fish water, embryos were transferred onto glass slides, set in 2% methylcellulose (Merck KGaA, Darmstadt, Germany), laterally oriented. Green fluorescent signals were acquired with an Axiozoom V16 fluorescence microscope (Zeiss) equipped with Axiocam 506 (Zeiss)and a digital camera. Apoptotic nuclei in trunk areas were analyzed and counted using ImageJ software.

### 2.9. Statistical Analysis

Statistical analysis was carried out using GraphPad Prism v8.0 (GraphPad Software, Inc., San Diego, CA, USA) software. Analysis of variance (ANOVA) followed by post hoc Bonferroni testing was used to determine the significant differences in the *lasp1* mRNA levels after morpholino injection and in the apoptosis assay. Data were considered statistically significant when *p*-values ≤ 0.05.

## 3. Results

### 3.1. Lasp1 Protein Is Expressed at Various Stages of Zebrafish Embryonic Development and in Several of Its Adult Organs

The zebrafish Lasp1 consists of 234 amino acids and shares sequence identity with human LASP1 (68%). LASP1 domains are conserved, including the cysteine-rich Lin11-Isl1-Mec3 (LIM) domain at the N-terminus (region 5–57), two Nebulin-repeats (NEBU, region 62–92 and 98–128) and the Src homology 3 (SH3) domain at the C-terminus (region 203–261). Each domain presents high sequence identity (80% for LIM, 86% and 97% NEBU domains, and 84% for SH3) (Appendix A). To investigate the role of Lasp1 in zebrafish development and organogenesis, we first determined its expression at different developmental stages and in adult organs. Western blotting analysis revealed Lasp1 expression in zebrafish at 48 and 72 h post-fertilization (hpf), at 6 days pf (Figure 1A, lanes 2–4) and in various adult organs, in particular, muscles, eyes, brains, livers and gastrointestinal tracts, where Lasp1 was slightly detectable (Figure 1A, lanes 5–10). Three-dimensional light-sheet microscopy analysis of the 72 hpf zebrafish specimens displayed Lasp1 expression in the trunk muscle fibers and also in a specific region of the yolk from which the liver originates (Figure 1B). Hematoxylin and eosin staining displayed the proper cell organization in the tissue sections of the examined organs (muscles, eyes, brains, livers and intestinal tracts) (Figure 2A); and immunohistochemistry carried out with Alexa photosensitive antibodies identified the localization and expression levels of Lasp1 in the same tissue sections. Lasp1 was expressed in the muscles, eyes, brain, liver and pancreas but was only slightly detectable in the intestine (Figure 2B).

### 3.2. Silencing of Lasp1 by Morpholino Leads to Morphology Alterations in Zebrafish Morphants

After assessing Lasp1 expression in zebrafish development and in different organs we started to study its function, performing knockdown experiments through *lasp1* specific morpholino (*lasp1* MO) injections. First, *lasp1* MO and STD MO were injected at 10 μM into zebrafish embryos at the one-cell stage, then at 48 h post-injection (48 hpi) we compared the morphology of the morphants following *lasp1* MO and standard MO (STD MO) injections. We classified the observed phenotypes of Lasp1 morphants as follows: normal phenotype (wild-type-like, WT-like), reduced head size and/or delay in development (mild phenotype), and underdeveloped body and/or undeveloped eyes (severe phenotype) (Figure 3A). Since MO injection may result in non-specific activation of p53 [21,22], we co-injected a p53 MO. The phenotypes induced by *lasp1* MO were similar to those obtained after co-injections of *lasp1* MO and p53 MO, suggesting that phenotypes induced by *lasp1* MO were not due to non-specific activation of p53 (Figure 3B). We then tested the effects of the *lasp1* MO at different doses (1, 10 and 100 µM). As shown in Figure 3C, most of the STD MO-injected embryos (96%) had normal phenotypes, while 4% of the morphants had the mild one. Most of the morphants obtained after 1 μM specific MO were WT-like (94%), and 6% showed the mild phenotype. On the contrary, morphants observed after 10 μM specific MO injections showed mainly altered phenotypes, with 32% and 43% showing the mild and severe phenotypes, respectively, and 25% the WT phenotype. Similarly, 23% and 70% of the morphants observed after 100 μM specific MO injections showed the mild and severe phenotypes, respectively, and 7% showed the normal one. We also observed that some of the morphants obtained after the specific MO injections were not alive: 5, 22 and 36% after 1, 10 and 100 μM MO injections; in the range of the expected results, 4 and 5% of the dead morphants observed in the control conditions received no MO injection and injections of ST MO, respectively. Based on these results, we decided to use the 10 μM dose of *lasp1* MO in the subsequent experiments. After discovering the specific perturbed phenotypes following *lasp1* MO injection in zebrafish one-cell-stage embryos, we ascertained reductions in expression levels of Lasp1 protein of 30%, 50% and 60% in the 1 μM, 10 μM and 100 μM *lasp1* MO samples (Figure 4), and the Lasp1 protein downmodulation was associated with increases in *lasp1* mRNA of 30, 66 and 89%, as determined by ddPCR (Figure 5). It is known that specific MO molecules targeting the ATG and/or splicing sites of the given mRNA molecules exert their action at the post-transcriptional level and that, if the targeted molecule is quite important in the given biological context, the cells can compensate the decreased levels of the protein by augmenting the corresponding mRNA expression [23].

### 3.3. Rescue of Zebrafish Phenotypes and Lasp1 Expression by Injection of LASP1 mRNA into the Zebrafish Embryos

In order to rescue the zebrafish phenotypes and Lasp1 expression, we co-injected zebrafish embryos at the one-cell stage with *lasp1* MO and *lasp1* MO + *LASP1* mRNA. As detailed in the Materials and Methods section, to obtain the *LASP1* transcript, we performed an in vitro transcription assay using an *LASP1*-expressing construct produced in our lab [9]. At 48 hpi, morphological phenotypes were observed, and protein extraction was performed to evaluate the expression levels of Lasp1 protein. As shown in Figure 6A, the co-injection of 10 μM *lasp1* MO and *lasp1* mRNA rescued Lasp1 protein expression, as detected by Western blotting (1.4-fold increase vs.10 μM *lasp1* MO). Concerning the phenotypes, most of the embryos co-injected with 10 μM *lasp1* MO + *LASP1* mRNA exhibited normal morphology (89%), and 6.5 and 4.5% displayed mild and severe phenotypes, respectively (Figure 6B). As expected, only 17% of the *lasp1* MO-injected embryo morphants exhibited the WT phenotype; most of the morphants exhibited the altered phenotype.

### 3.4. Lasp1 Silencing by MO Leads to an Increase in Apoptosis

Since LASP1 plays a role in favouring cell proliferation and the migration of human cancer cells and few data on apoptosis have been reported [24,25,26,27,28], we tested the effects of Lasp1 silencing on apoptosis to explore its biological role in zebrafish embryonic development. As expected, basal programmed cell death was recorded both in STD MO-injected embryos and in those not injected (Figure 7); for the first time, the findings obtained displayed an increase in cell death following *lasp1* MO treatment, reaching a fold increase (FI) of 1.9 after 100 μM *lasp1* MO injection (Figure 7B). As detailed in the Materials and Methods section, the assay used to examine apoptosis was the acridine orange assay that selectively stains the apoptotic forms of death [29].

## 4. Discussion

The overexpression of LASP1 in different types of human cancer is associated with unfavourable prognoses for cancer patients, and RNAi knock-down of LASP1 has led to strong inhibition of the proliferation and migration of various cancer cell lines. LASP1 has been initially characterized as a cytoskeletal protein, and more recently some studies have ascertained the ubiquitous expression of its mRNA in most human tissues and organs. Further, some data have revealed high expression level of LASP1 in the fetal human brain and liver, opening the field of LASP1 in embryogenesis. In the current study, we used a zebrafish model to address the role of Lasp1 in embryonic development because it is a powerful model in this scientific field, especially due to its rapid development and the transparency of its developing embryos, which aids the observation of morphological alterations. Our work presents new basic knowledge on Lasp1. For the first time, we determined Lasp1 expression in zebrafish embryonic development and in several organs of zebrafish adults. Further, we tested the function of Lasp1 by the morpholino (MO) approach, monitoring the abnormal morphants obtained following Lasp1 knockdown, and we also studied in these conditions the biological effects on apoptosis. The results obtained by Western blotting clearly showed Lasp1 expression during zebrafish embryogenesis, at 48 and 72 hpf, as well as at 6 days pf, and light-sheet microscopy fluorescence was used to visualize Lasp1 expression in trunk somites and livers. For the first time, we presented the normal organizations of several tissues in zebrafish adults detected by H&E staining, with the corresponding Lasp1 expressions visualized by immunohistochemistry carried out with Alexa photosensitive antibodies. Concerning the fact that, in our Western blotting experiments, zebrafish and human LASP1 were revealed at 35 and 37 kDa, respectively, it is necessary to mention that human LASP1 is a protein of 261 amino acids with a molecular mass of 29.7 that runs in SDS-PAGE at 37 kDa and with no structural explanation. Similarly, in our experiments, zebrafish Lasp1 (a protein with a 75% similarity to human LASP1, consisting of 234 amino acids and with a predicted molecular mass of 25.6 kDa) ran at 35 kDa. After ascertaining Lasp1 expression in embryonic development and in zebrafish adults, we tested its function by the MO approach, which has been successfully used in studies on zebrafish for over 20 years [30,31,32]. Specific *lasp1* MO led mostly to morphants with abnormal phenotypes, which were recorded 48 h after injection at the one-cell stage and were not due to non-specific activation of p53; *lasp1* MO led to a decrease in Lasp1 protein expression in the range of 30–60% when different doses of *lasp1* MO were used (1, 10 and 100 μM). These data strongly suggest a major role for Lasp1 in zebrafish embryonic development, since the perturbed morphologies of the abnormal morphants are various and seem to involve several organs. We also determined, by ddPCR, the expression of *lasp1* mRNA, which increased after *lasp1* MO injection. As already stated by some authors [23], it may be that a specific MO can decrease the protein expression of the target gene while embryo cells might compensate the loss of protein enhancement by activating the transcription of the target. This intriguing interpretation deserves a future study in order to determine the molecular mechanisms of *lasp1* expression regulation in embryonic development. Being aware that LASP1 favours the process of cell proliferation, at least in human cancer cell lines in vitro, and that it may play a role in reducing cell apoptosis rates in certain contexts [24,25,26,27,28], we started exploring the biological effects of *lasp1* MO in zebrafish development, looking at the process of apoptosis, which increased 48 h after *lasp1* MO injection at the one-cell stage. Probably, the loss of Lasp1 protein may affect the ability of cells to proliferate, favouring apoptosis. In conclusion, from a general point of view, the present work is the first effort to study Lasp1 expression and its role in zebrafish embryonic development; it makes an original contribution to basic knowledge and will serve to promote further studies aiming to understand the role of human and zebrafish LASP1 in physiological conditions.

In summary, in the current study, we have presented novel data on protein Lasp1 expression in various organs of the adult zebrafish organism as well as in its embryonic development at 48 and 72 h and 6 days post-fertilization. We have described an effective use of antisense morpholino oligonucleotides to target the translation of *lasp1* mRNA and have ascertained that the effects of *lasp1* MO are specific and lead to morphants with perturbated morphologies and various phenotypes. These biological effects on the morphants, as well as the increase in apoptosis, occur during the embryonic development of zebrafish. Since this model organism is one of the best for studying the development of vertebrates and to test the function of a given gene, our data support the use of zebrafish to deepen understanding of the biological function of Lasp1.

## Figures and Tables

**Figure 1 genes-14-00035-f001:**
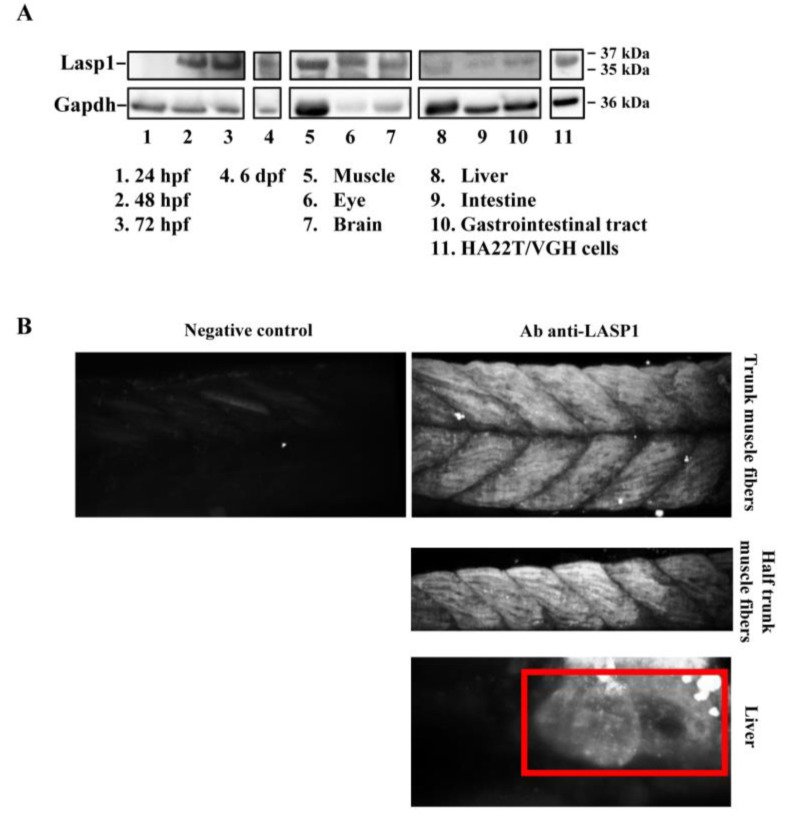
Detection of Lasp1 protein expression in zebrafish. (**A**) Western blotting analysis of Gapdh and Lasp1 at different embryonic stages (24 hpf, 48 hpf, 72 hpf and 6 dpf) and in different tissues from zebrafish adults (muscle, eye, brain, liver, intestine, and gastrointestinal tract tissues). Human hepatocellular carcinoma cells, HA22T/VGH, with high levels of LASP1, were used as positive controls. (**B**) Localization of Lasp1 evaluated using light-sheet fluorescence microscopy in zebrafish at 72 hpf. Overview of muscle fibers and liver. Negative control is a representative overview of the “trunk muscle fibers” without primary anti-Lasp1 antibodies and with the fluorescent-labeled secondary antibodies. The images are representative of at least two independent experiments.

**Figure 2 genes-14-00035-f002:**
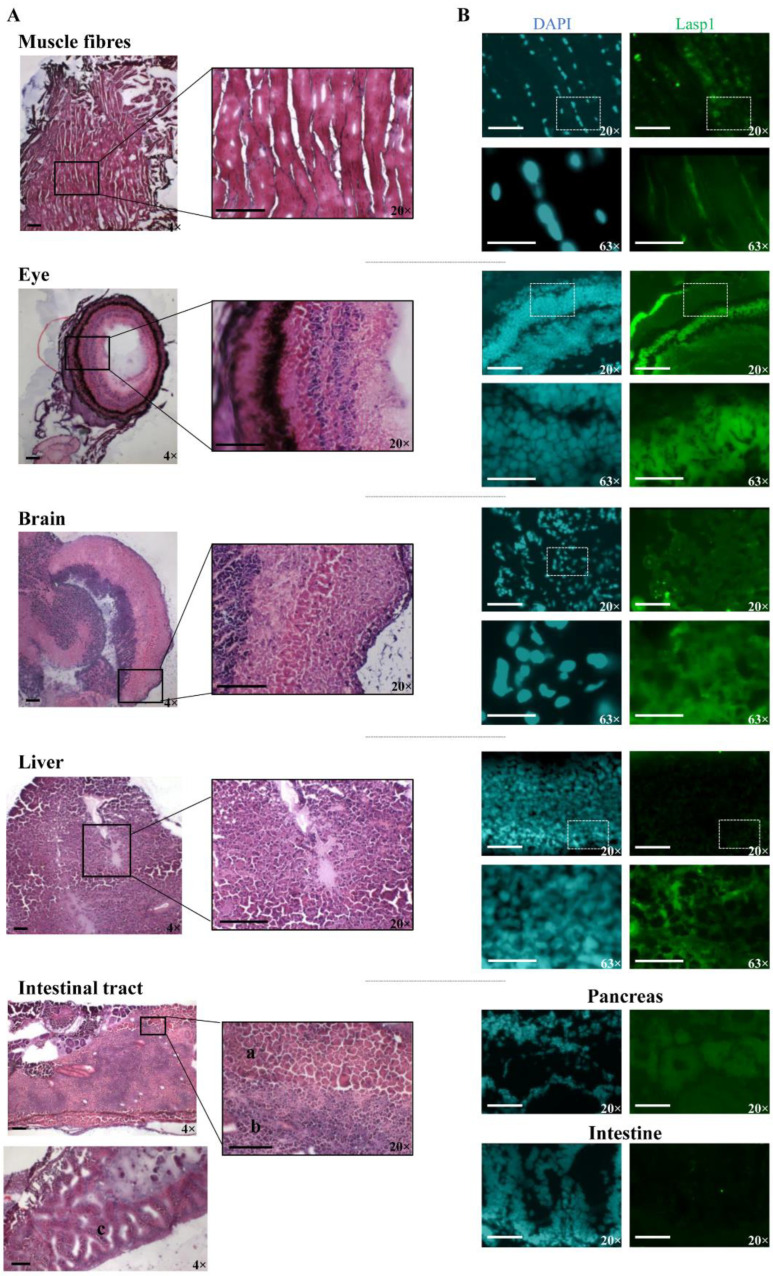
Detection of Lasp1 protein expression in various organs and tissues of zebrafish, including the muscle, the eye, the brain, the liver and the gastrointestinal tract. (**A**) Histological analysis by hematoxylin and eosin staining of muscle fiber, eye, brain, liver and intestinal tract tissue sections. Pancreas (**a**), liver (**b**), and intestine (**c**) samples are indicated. Magnification: 4× and 20×. (**B**) Immunofluorescence detection of Lasp1 (green) in the selected tissue sections by immunohistochemistry with Alexa photosensitive antibodies. Magnification: 20× and 63× (blue: DAPI; green: FITC). Scale bars are placed at the lower left corner of each image and correspond to 100 µm for 4× and 20× magnification and to 30 µm for 63× magnification.

**Figure 3 genes-14-00035-f003:**
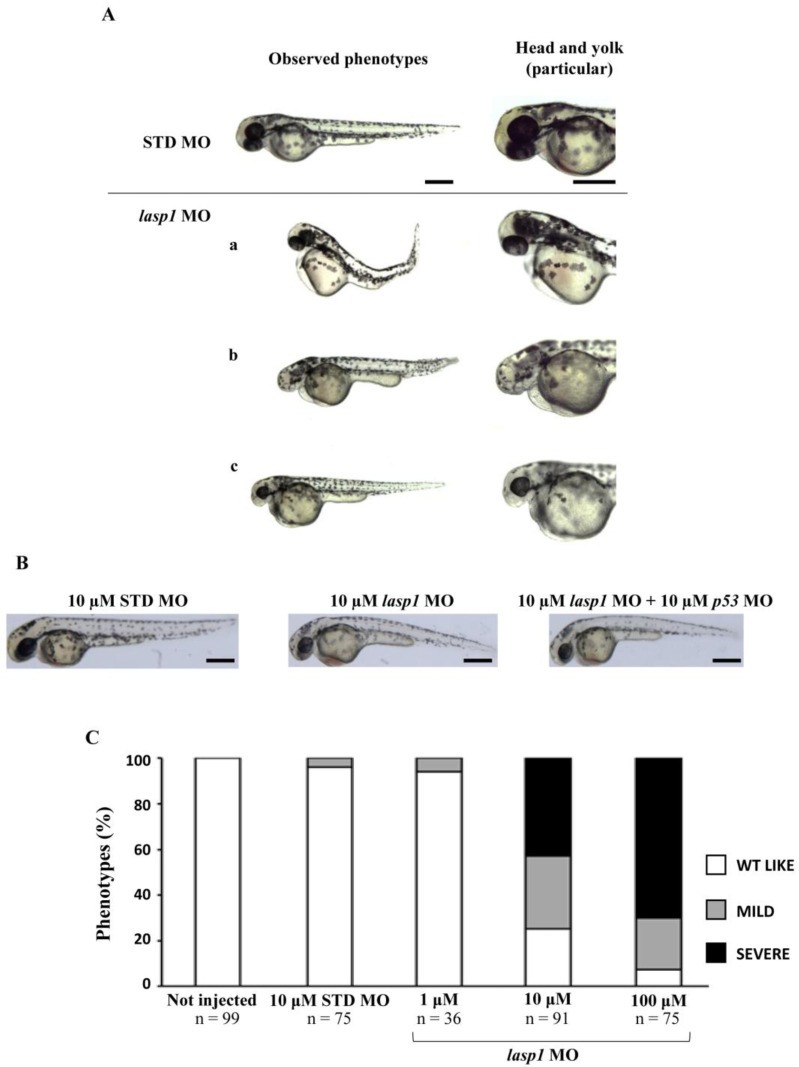
The evaluation of the morphological defects in *lasp1* MO morphants. (**A**) Representative images of the phenotypes of zebrafish embryos injected with 10 µM control morpholino (STD MO) or 10 µM *lasp1* MO and photographed at 48 hpf. Morphants exhibited different morphological defects, including the severe phenotype with underdeveloped body and heart edema (**a**); undeveloped eyes and reduced tail size (**b**); and the mild phenotype with reduced head size, hypopigmentation and thick yolk extension (**c**). (**B**) No differences were observed between embryos injected with *lasp1*-MO alone and those injected with the combination of *lasp1* MO and p53 MO. Scale bars: 200 µm. (**C**) Analysis of morphological phenotypes in *lasp1* MO morphants revealed developmental defects. Most of the 10 µM and 100 µM *lasp1* 1 MO morphants revealed mild or severe phenotypes; the ST MO and the 1 µM *lasp1* MO morphants mainly showed the wild-type phenotype. The number under each bar is the total number of embryos examined under each experimental condition. Three biological replicates were used for each assay.

**Figure 4 genes-14-00035-f004:**
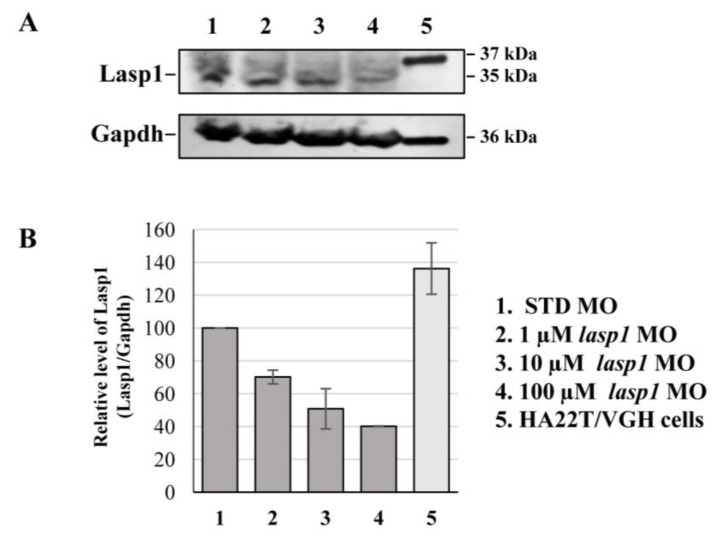
Detection of the decreased expression of Lasp1 protein in *lasp1* MO morphants. (**A**) Representative Western blotting for Lasp1 protein level in 48 hpf embryos injected with STD MO or *lasp1* MO at different doses (1, 10 and 100 µM). Human hepatocellular carcinoma cells (HA22T/VGH) were used as positive controls for LASP1 expression. Gapdh was used as an internal control. The images are representative of two independent experiments. (**B**) Relative quantifications of Lasp1 expression levels. The histograms represent the mean IOD (integrated optical density) values in percentages; the bars represent SDs from two independent experiments.

**Figure 5 genes-14-00035-f005:**
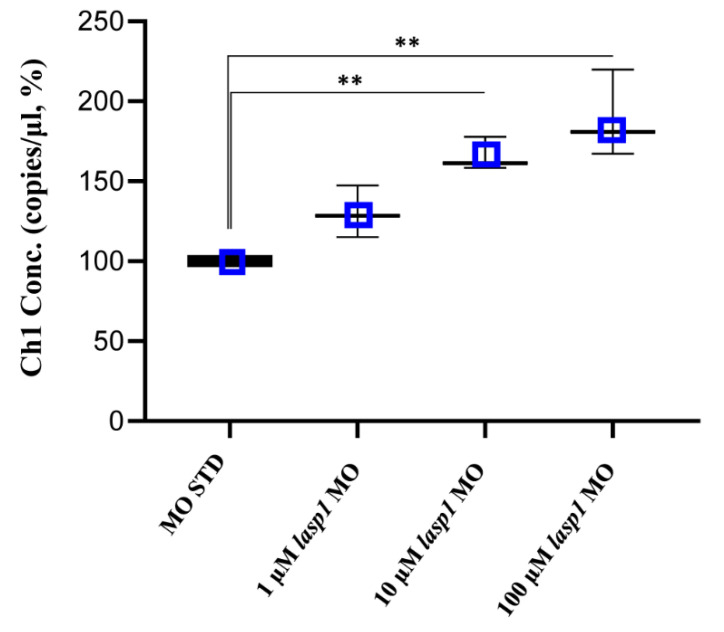
Absolute quantifications of *lasp1* mRNA in *lasp1* MO morphants using ddPCR. Dose-dependent increases in *lasp1* mRNA levels were observed. The graph represents the average values of copies/μL referred to the MO STD samples; error bars are SDs. ** *p* < 0.01 in one-way ANOVA followed by Bonferroni testing.

**Figure 6 genes-14-00035-f006:**
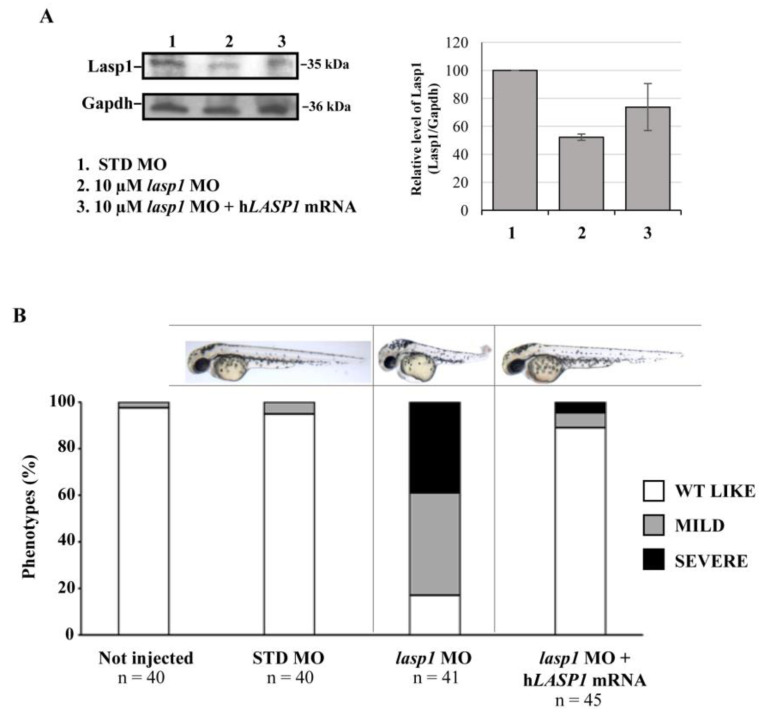
Co-injection of *lasp1* MO and *LASP1* mRNA rescues the morphological defects of morphants. (**A**) Representative Western blotting and relative quantifications of Lasp1 protein levels in 48 hpf embryos injected with STD MO, 10 µM *lasp1* MO or co-injected with 10 µM *lasp1* MO and LASP1 mRNA (50 ng). GAPDH was used as an internal control. The histograms represent the mean IOD (integrated optical density) values in percentages; the bars are SDs. (**B**) Analysis of phenotypes and corresponding representative images of the zebrafish embryos injected with 10 µM control morpholino (STD MO), 10 µM *lasp1* MO or co-injected with 10 µM *lasp1* MO and human *LASP1* mRNA (50 ng) and photographed at 48 hpf. The number under each bar is the total number of embryos examined under each experimental condition. Three biological replicates were used for each assay.

**Figure 7 genes-14-00035-f007:**
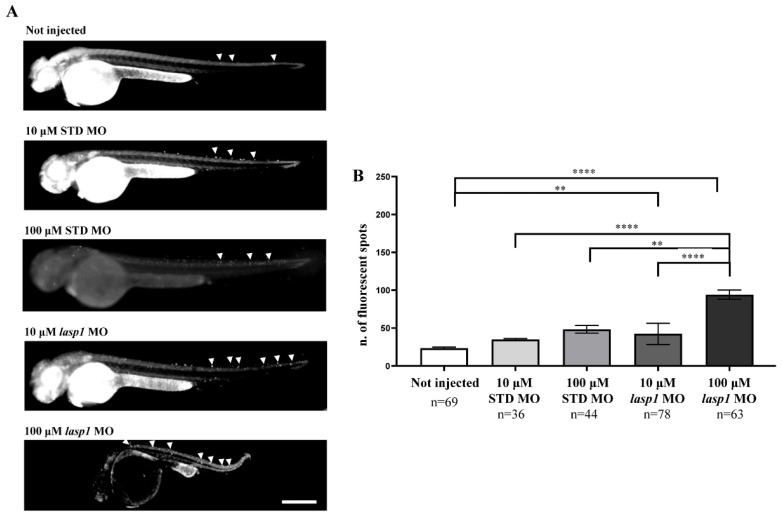
Injection of *lasp1* MO increases apoptosis in zebrafish embryos at 48 hpf. (**A**) Representative images of acridine orange staining by fluorescence microscopy show increased apoptosis in whole embryos following injections of *lasp1* MO at 10 µM and 100 µM doses. Fluorescence spots are evidenced by arrowheads. Scale bars: 200 µm. (**B**) The acridine orange-positive spots were counted using ImageJ. Histograms represent the average numbers of fluorescent spots; error bars are SEMs. ** *p* < 0.01 and **** *p* < 0.0001 in one-way ANOVA followed by Bonferroni testing.

## Data Availability

The data supporting the findings of this study are contained within the contents of this article. The datasets generated during this study will be freely provided by the corresponding author upon request.

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
