# Peer review of "Lasp1 Expression Is Implicated in Embryonic Development of Zebrafish"

_genes, 2022, doi:10.3390/genes14010035_

Round 1

Reviewer 1 Report

The first part of the paper is mainly descriptive, analysing the expression of LASP1 orthologue in zebrafish adult tissues. What about the conservation of index of the protein, a part from the information included in the Discussion? It could be useful to perform aminoacidic sequence alignment and/or analysis using software such as  DIOPT Ortholog Finder (https://www.flyrnai.org/cgi-bin/DRSC_orthologs.pl), to analyse the conservation of the different domains. Please take care of the correct nomenclature for zebrafish gene/mRNA (lasp1 in italics) and protein (Lasp1).

Figure 1A. In Western blot analysis the evaluation of an housekeeping gene expression should be included.

Using EMBL Expression Atlas (https://www.ebi.ac.uk/gxa/experiments/E-ERAD-475/Results), lasp1 mRNA seems highly expressed during the early stages of zebrafish development, suggesting a maternal origin of the transcript. Did the author check the expression of the protein before 48 hpf?

Lines 197-198. The sentence “In the muscle LASP1 had both cytoplasmic and nuclear localization (Figure  2B)” is not supported by the results showed: a merge image is missing and in some cases the DAPI signal seems saturated, especially in the case of the images taken with the 63X objective. If available, it would be better to include less saturated images.

Regarding the experiments with morpholinos, in the “Materials and methods” section the authors declared to use a mixture of two different sequences, targeting lasp1-201 and lasp1-202. However, the authors did not verify the expression of the two mRNA isoforms during the early stage of zebrafish development. In the validation of lasp1 knock-down by morpholino exposure (Figure 5), do the authors use primers amplifying the cDNA derived from both the isoforms? Since they found an increase of mRNA level, a compensatory effect can be present (as also expected by the authors given the comment at line 350-351), so it seems important to understand the contribution of the two single mRNAs targeted by morpholinos.

Figure 3C. Statistical analysis is missing as well as the number of embryos analysed.

Figure 4, Figure 6A. The authors showed a densitometric analysis of Western blot bands but the number of experiments performed has not been indicated.

Figure 6B. The number of embryos analysed is missing.

Regarding the apoptosis evaluation, it not surprising that a such altered lasp1 morphant embryo present high levels of cell death at the dose of 100 uM. It would be a consequence of the altered development induced by lasp1 morpholino. The use of the lower dose of morpholino is more convincing and moreover the proper negative control group using the same dose of STD morpholino is included. If possible, the authors might consider to include a positive control in the experimental design, such as a treatment inducing apoptosis evaluate by acridine orange staining. Again, the number of embryos analysed is missing.

Line 353. “LASP1 favours the process of cell proliferation”, so why the authors focused on apoptosis and not on proliferation?

Minor points

In the results paragraph “2.8. Acridine orange assay for apoptosis evaluation” the authors referred to “GFP” the acronym for Green Fluorescent Protein. Maybe they meant just the green fluorescent signal.

Line 213. Does “green- FITCH” mean “green-FITC”?

Figure 2B. Please explain in the panel of the Intestinal tract what do the letter “a”, “b” and “c” mean.

Figure 3A and Figure 3B. Scale bars are missing.

Author Response

The first part of the paper is mainly descriptive, analysing the expression of LASP1 orthologue in zebrafish adult tissues. What about the conservation of index of the protein, a part from the information included in the Discussion? It could be useful to perform aminoacidic sequence alignment and/or analysis using software such as DIOPT Ortholog Finder (https://www.flyrnai.org/cgi-bin/DRSC_orthologs.pl), to analyse the conservation of the different domains.

We thank the reviewer for this suggestion. In the revised version of the manuscript we have included the supplementary figure 1 (Fig. S1) reporting the aminoacidic sequence alignment by using the software DIOPT Ortholog Finder.

 Please take care of the correct nomenclature for zebrafish gene/mRNA (lasp1 in italics) and protein (Lasp1).

We have modified the nomenclature for zebrafish gene/mRNA protein according to the reviewer’s request.

Figure 1A. In Western blot analysis the evaluation of a housekeeping gene expression should be included.

We have included in Figure 1 the Gapdh protein expression analysis performed by WB 

Using EMBL Expression Atlas (https://www.ebi.ac.uk/gxa/experiments/E-ERAD-475/Results), lasp1 mRNA seems highly expressed during the early stages of zebrafish development, suggesting a maternal origin of the transcript.

We thank the reviewer for this comment and we agree.

Did the author check the expression of the protein before 48 hpf?

We checked Lasp1 protein expression at 24 hpf. We added this result in the Fig. 1A

Lines 197-198. The sentence “In the muscle LASP1 had both cytoplasmic and nuclear localization (Figure 2B)” is not supported by the results showed: a merge image is missing and in some cases the DAPI signal seems saturated, especially in the case of the images taken with the 63X objective. If available, it would be better to include less saturated images.

We agree with the reviewer. We removed the sentence reported in lines 197-198. Anyway we provided a less saturated image.

Regarding the experiments with morpholinos, in the “Materials and methods” section the authors declared to use a mixture of two different sequences, targeting lasp1-201 and lasp1-202. However, the authors did not verify the expression of the two mRNA isoforms during the early stage of zebrafish development. In the validation of lasp1 knock-down by morpholino exposure (Figure 5), do the authors use primers amplifying the cDNA derived from both the isoforms? Since they found an increase of mRNA level, a compensatory effect can be present (as also expected by the authors given the comment at line 350-351), so it seems important to understand the contribution of the two single mRNAs targeted by morpholinos. We thank the reviewer for this observation. We have included in the revised version of the manuscript a new supplementary Figure 2 (Fig. S2) that clarifies this aspect. The hydrolysis probe-based assay used to quantify the expression levels of lasp1 mRNA by ddPCR detected both the mRNA isoforms lasp1-210 and lasp1-202. For this reason, although the assay is indicated by Thermofisher as “best coverage” (assay ID: Dr03439091_m1), it does not distinguish the contribution of the two single mRNAs targeted by morpholinos.

Figure 3C. Statistical analysis is missing as well as the number of embryos analysed.

In Figure 3 of the revised version of the manuscript we added the number of embryos analysed. We decided to calculate the percentage relative to the total number of embryos analysed in all 3 experiments and for each phenotype. Thus, for each phenotype class a single percentage value was obtained.

Figure 4, Figure 6A. The authors showed a densitometric analysis of Western blot bands but the number of experiments performed has not been indicated.

We have reported the number of the experiments performed in the correspondent figure legends

Figure 6B. The number of embryos analysed is missing.

In Figure 6B the number of embryos analysed has been added.

Regarding the apoptosis evaluation, it not surprising that a such altered lasp1 morphant embryo present high levels of cell death at the dose of 100 uM. It would be a consequence of the altered development induced by lasp1 morpholino. The use of the lower dose of morpholino is more convincing and moreover the proper negative control group using the same dose of STD morpholino is included. If possible, the authors might consider to include a positive control in the experimental design, such as a treatment inducing apoptosis evaluate by acridine orange staining. Again, the number of embryos analysed is missing.

We thank the Referee for these comments. We have added in Fig.7 the results obtained treating the zebrafish embryos with 100mM ST MO; as requested we have also reported the number of embryos analysed. As suggested by the Referee we considered a positive control of apoptosis and we treated the zebrafish embryos with rotenone, but the green dots were too many and uncountable. For your convenience, we include below a representative image of the positive control of apoptosis.

Line 353. “LASP1 favours the process of cell proliferation”, so why the authors focused on apoptosis and not on proliferation?

Being aware that LASP1 favours the process of cell proliferation, at least in human cancer cell lines in vitro, and   it may play a role in reducing the cell apoptosis rate in certain contexts, we decided to start in exploring the biological effects of lasp1 MO in zebrafish development looking at the process of apoptosis that resulted to be increased 48 hours after lasp1 MO injection at one cell stage. In a future work we will elaborate deeply the role of Lasp1 in other biological processes such as proliferation, migration and invasion. As requested by Referee n.2 we have added some relevant references on LASP1 knockdown that increases apoptosis in human cancer cell lines.

Minor points

In the results paragraph “2.8. Acridine orange assay for apoptosis evaluation” the authors referred to “GFP” the acronym for Green Fluorescent Protein. Maybe they meant just the green fluorescent signal.

Yes, we apologize for this error. We meant the green fluorescent signal. We have corrected this information in the text.

Line 213. Does “green- FITCH” mean “green-FITC”?

Yes it does, we apologize for this typing error.

Figure 2B. Please explain in the panel of the Intestinal tract what do the letter “a”, “b” and “c” mean.

They mean (a) pancreas, (b) liver, (c) intestine. It is written in the legend of Figure 2.

Figure 3A and Figure 3B. Scale bars are missing.

In the revised version of the manuscript we have reported the scale bars.

The first part of the paper is mainly descriptive, analysing the expression of LASP1 orthologue in zebrafish adult tissues. What about the conservation of index of the protein, a part from the information included in the Discussion? It could be useful to perform aminoacidic sequence alignment and/or analysis using software such as DIOPT Ortholog Finder (https://www.flyrnai.org/cgi-bin/DRSC_orthologs.pl), to analyse the conservation of the different domains.

We thank the reviewer for this suggestion. In the revised version of the manuscript we have included the supplementary figure 1 (Fig. S1) reporting the aminoacidic sequence alignment by using the software DIOPT Ortholog Finder.

 Please take care of the correct nomenclature for zebrafish gene/mRNA (lasp1 in italics) and protein (Lasp1).

We have modified the nomenclature for zebrafish gene/mRNA protein according to the reviewer’s request.

Figure 1A. In Western blot analysis the evaluation of a housekeeping gene expression should be included.

We have included in Figure 1 the Gapdh protein expression analysis performed by WB 

Using EMBL Expression Atlas (https://www.ebi.ac.uk/gxa/experiments/E-ERAD-475/Results), lasp1 mRNA seems highly expressed during the early stages of zebrafish development, suggesting a maternal origin of the transcript.

We thank the reviewer for this comment and we agree.

Did the author check the expression of the protein before 48 hpf?

We checked Lasp1 protein expression at 24 hpf. We added this result in the Fig. 1A

Lines 197-198. The sentence “In the muscle LASP1 had both cytoplasmic and nuclear localization (Figure 2B)” is not supported by the results showed: a merge image is missing and in some cases the DAPI signal seems saturated, especially in the case of the images taken with the 63X objective. If available, it would be better to include less saturated images.

We agree with the reviewer. We removed the sentence reported in lines 197-198. Anyway we provided a less saturated image.

Regarding the experiments with morpholinos, in the “Materials and methods” section the authors declared to use a mixture of two different sequences, targeting lasp1-201 and lasp1-202. However, the authors did not verify the expression of the two mRNA isoforms during the early stage of zebrafish development. In the validation of lasp1 knock-down by morpholino exposure (Figure 5), do the authors use primers amplifying the cDNA derived from both the isoforms? Since they found an increase of mRNA level, a compensatory effect can be present (as also expected by the authors given the comment at line 350-351), so it seems important to understand the contribution of the two single mRNAs targeted by morpholinos. We thank the reviewer for this observation. We have included in the revised version of the manuscript a new supplementary Figure 2 (Fig. S2) that clarifies this aspect. The hydrolysis probe-based assay used to quantify the expression levels of lasp1 mRNA by ddPCR detected both the mRNA isoforms lasp1-210 and lasp1-202. For this reason, although the assay is indicated by Thermofisher as “best coverage” (assay ID: Dr03439091_m1), it does not distinguish the contribution of the two single mRNAs targeted by morpholinos.

Figure 3C. Statistical analysis is missing as well as the number of embryos analysed.

In Figure 3 of the revised version of the manuscript we added the number of embryos analysed. We decided to calculate the percentage relative to the total number of embryos analysed in all 3 experiments and for each phenotype. Thus, for each phenotype class a single percentage value was obtained.

Figure 4, Figure 6A. The authors showed a densitometric analysis of Western blot bands but the number of experiments performed has not been indicated.

We have reported the number of the experiments performed in the correspondent figure legends

Figure 6B. The number of embryos analysed is missing.

In Figure 6B the number of embryos analysed has been added.

Regarding the apoptosis evaluation, it not surprising that a such altered lasp1 morphant embryo present high levels of cell death at the dose of 100 uM. It would be a consequence of the altered development induced by lasp1 morpholino. The use of the lower dose of morpholino is more convincing and moreover the proper negative control group using the same dose of STD morpholino is included. If possible, the authors might consider to include a positive control in the experimental design, such as a treatment inducing apoptosis evaluate by acridine orange staining. Again, the number of embryos analysed is missing.

We thank the Referee for these comments. We have added in Fig.7 the results obtained treating the zebrafish embryos with 100mM ST MO; as requested we have also reported the number of embryos analysed. As suggested by the Referee we considered a positive control of apoptosis and we treated the zebrafish embryos with rotenone, but the green dots were too many and uncountable. For your convenience, we include below a representative image of the positive control of apoptosis.

Line 353. “LASP1 favours the process of cell proliferation”, so why the authors focused on apoptosis and not on proliferation?

Being aware that LASP1 favours the process of cell proliferation, at least in human cancer cell lines in vitro, and   it may play a role in reducing the cell apoptosis rate in certain contexts, we decided to start in exploring the biological effects of lasp1 MO in zebrafish development looking at the process of apoptosis that resulted to be increased 48 hours after lasp1 MO injection at one cell stage. In a future work we will elaborate deeply the role of Lasp1 in other biological processes such as proliferation, migration and invasion. As requested by Referee n.2 we have added some relevant references on LASP1 knockdown that increases apoptosis in human cancer cell lines.

Minor points

In the results paragraph “2.8. Acridine orange assay for apoptosis evaluation” the authors referred to “GFP” the acronym for Green Fluorescent Protein. Maybe they meant just the green fluorescent signal.

Yes, we apologize for this error. We meant the green fluorescent signal. We have corrected this information in the text.

Line 213. Does “green- FITCH” mean “green-FITC”?

Yes it does, we apologize for this typing error.

Figure 2B. Please explain in the panel of the Intestinal tract what do the letter “a”, “b” and “c” mean.

They mean (a) pancreas, (b) liver, (c) intestine. It is written in the legend of Figure 2.

Figure 3A and Figure 3B. Scale bars are missing.

In the revised version of the manuscript we have reported the scale bars.

Reviewer 2 Report

Zebrafish is an excellent tool for studying gene functions, particularly in developmental biology. Grossi and co-authors investigated LASP1 for the first time in zebrafish by examining its expression at various stages during embryonic development and in adults. They studied LASP1 functions by analyzing phenotypes in LASP1 MO morphants and found an increased apoptosis frequency in LASP1 MO. The text is well-written, and the figures are well-constructed. I read the manuscript with great interest. However, I have the following questions that I hope the authors can address before its publication in Genes.

  1. Fig 1A, add loading control (GAPDH?). It is not clear what the control in Fig 1B is. Please explain in method and in figure legend.  
  2. Add scale bar in Fig2. Negative control images for staining should be provided either in Fig2 or as a supplemental figure for better interpretations of the IHC/IF results. 
  3. Please provide references for MO-induced nonspecific p53 activation. 
  4. For figures that contain quantification (Fig 3, 4, 6, 7), please provide the number of embryos being analyzed in each condition and the number of independent experiments performed. 
  5. LASP1 knockdown increasing apoptosis in cancerous cell lines has been reported by several in the literature. Please add relevant citations.
  6. Please provide the clone number for monoclonal antibodies. The clone number is helpful information for those who wish to replicate the experiment, as the vendors may have other clones for the same antibody over time.  

Author Response

Zebrafish is an excellent tool for studying gene functions, particularly in developmental biology. Grossi and co-authors investigated LASP1 for the first time in zebrafish by examining its expression at various stages during embryonic development and in adults. They studied LASP1 functions by analyzing phenotypes in LASP1 MO morphants and found an increased apoptosis frequency in LASP1 MO. The text is well-written, and the figures are well-constructed. I read the manuscript with great interest.

We thank the reviewer for appreciating our work and the manuscript.

However, I have the following questions that I hope the authors can address before its publication in Genes.

  1. Fig 1A, add loading control (GAPDH?).

In the revised version of the manuscript we have reported WB analysis of the expression levels of Gapdh protein.

It is not clear what the control in Fig 1B is. Please explain in method and in figure legend.  

In Figure 1B the control is a representative overview of the “trunk muscle fibers” without the primary antibodies anti-lasp1 and with fluorescent-labeled secondary antibodies. We have explained in methods and in the legend of figure 2B, as requested.

Add scale bar in Fig2.

  1. Done, as also requested by reviewer n. 1.

Negative control images for staining should be provided either in Fig2 or as a supplemental figure for better interpretations of the IHC/IF results. 

We have included a new supplementary Figure (Fig. S1) with the negative control images for each tissue section reported in Fig. 2.

  1. Please provide references for MO-induced nonspecific p53 activation. 

The following references have been added and cited in the text:

  1. a) Mara E Robu Jon D Larson, Aidas Nasevicius, Soraya Beiraghi, Charles Brenner, Steven A Farber, Stephen C Ekker.

p53 activation by knockdown technologies.

2007 May 25;3(5):e78. doi: 10.1371/journal.pgen.0030078.

  1. b) Sebastian S Gerety and David G Wilkinson. Morpholino artifacts provide pitfalls and reveal a novel role for pro-apoptotic genes in hindbrain boundary development. 

2011 Feb 15;350(2):279-89. doi: 10.1016/j.ydbio.2010.11.030.

  1. For figures that contain quantification (Fig 3, 4, 6, 7), please provide the number of embryos being analyzed in each condition and the number of independent experiments performed. 

The number of embryos analysed has been added and we have reported the number of the experiments performed.

  1. LASP1 knockdown increasing apoptosis in cancerous cell lines has been reported by several in the literature. Please add relevant citations.

Done. We have included and cited the following references:

LIM and SH3 protein 1 regulates cell growth and chemosensitivity of human glioblastoma via the PI3K/AKT pathway  Zhong C, Chen Y, Tao B, Peng L, Peng T, Yang X, Xia X, Chen L.

BMC Cancer. 2018 Jul 6;18(1):722. doi: 10.1186/s12885-018-4649-2.  PMID: 29980193

LASP1 promotes nasopharyngeal carcinoma progression through negatively regulation of the tumor suppressor PTEN.  Gao Q, Tang L, Wu L, Li K, Wang H, Li W, Wu J, Li M, Wang S, Zhao L.

Cell Death Dis. 2018 Mar 12;9(3):393. doi: 10.1038/s41419-018-0443-y.  PMID: 29531214

Role of LASP-1, a novel SOX9 transcriptional target, in the progression of lung cancer

Jianguang Shi 1, Jing Guo 1, Xinjian Li 1   Int J Oncol   2018 Jan;52(1):179-188. doi:10.3892/ijo.2017.4201. PMID: 29138807

MicroRNA-218 inhibits the proliferation, migration, and invasion and promotes apoptosis of gastric cancer cells by targeting LASP1.  Wang LL, Wang L, Wang XY, Shang D, Yin SJ, Sun LL, Ji HB.

Tumour Biol. 2016 Nov;37(11):15241-15252. doi: 10.1007/s13277-016-5388-0. PMID: 27696291 

miR-133 inhibits proliferation and promotes apoptosis by targeting LASP1 in lupus nephritis

Zhimin Huang 1, Guozhen Pang 2, Yu Ge Huang 1, Chengyan Li 1

Exp Mol Pathol 2020 Jun;114:104384. doi: 10.1016/j.yexmp.2020.104384. PMID: 31987844.

  1. Please provide the clone number for monoclonal antibodies. The clone number is helpful information for those who wish to replicate the experiment, as the vendors may have other clones for the same antibody over time.  

Done. In Materials and Methods section we have included the clone number. Mouse anti-human LASP1 monoclonal antibody, clone 8C6 (Millipore).

Round 2

Reviewer 1 Report

The revised version of the manuscript included modifications and corrections that answer to reviewer's concerns in a satisfying manner.

Reviewer 2 Report

I appreciate the authors making an effort to address my comments. All my concerns have been appropriately addressed.